# Effect of Low-Level Laser Therapy on Periodontal Host Cells and a Seven-Species Periodontitis Model Biofilm

**DOI:** 10.3390/ijms26146803

**Published:** 2025-07-16

**Authors:** Selma Dervisbegovic, Susanne Bloch, Vera Maierhofer, Christian Behm, Xiaohui Rausch-Fan, Andreas Moritz, Christina Schäffer, Oleh Andrukhov

**Affiliations:** 1Division of Periodontology, University Clinic of Dentistry, Medical University of Vienna, 1090 Vienna, Austria; selma.dervisbegovic@meduniwien.ac.at; 2Competence Center for Periodontal Research, University Clinic of Dentistry, Medical University of Vienna, 1090 Vienna, Austria; susanne.bloch@meduniwien.ac.at (S.B.); vera.maierhofer@meduniwien.ac.at (V.M.); christian.behm@meduniwien.ac.at (C.B.); 3NanoGlycobiology Research Group, Institute of Biochemistry, Department of Natural Sciences and Sustainable Resources, University of Natural Resources and Life Sciences, 1190 Vienna, Austria; christina.schaeffer@boku.ac.at; 4Center for Clinical Research, University Clinic of Dentistry, Medical University of Vienna, 1090 Vienna, Austria; xiaohui.rausch-fan@meduniwien.ac.at; 5Clinical Division of Conservative Dentistry, University Clinic of Dentistry, Medical University of Vienna, 1090 Vienna, Austria; andreas.moritz@meduniwien.ac.at

**Keywords:** low level laser therapy, seven-species biofilms, inflammatory mediators, gingival fibroblasts, periodontal ligament cells

## Abstract

Low-level laser therapy (LLLT) is gaining attention as an effective adjunct to non-surgical periodontal treatment. This study evaluates the potential of LLLT to reduce bacterial load in a clinically relevant in vitro subgingival biofilm model and its impact on the inflammatory response. A subgingival biofilm model consisting of seven bacterial species was established. Primary human gingival fibroblasts (GFs) and periodontal ligament cells (PDLs) were cultured. Both biofilms and host cells were treated with the DenLase Diode Laser (980 nm) under various clinically relevant settings. The composition and structure of the seven-species biofilms were evaluated using quantitative PCR and fluorescence microscopy, respectively. The inflammatory response in host cells was analyzed by measuring the gene and protein expression levels of various inflammatory mediators. Laser treatment at power outputs ranging from 0.3 to 2 W had no significant effect on biofilm composition or architecture. LLLT, particularly at higher power settings, reduced the viability in both GFs and PDLs up to 70%. Gene expression levels of inflammatory mediators were only minimally influenced by laser treatment. However, LLLT significantly decreased the secretion of all examined cytokines. These findings suggest that LLLT with a 980 nm diode laser, under clinically relevant conditions, exerts anti-inflammatory rather than antimicrobial effects.

## 1. Introduction

Periodontitis is a chronic inflammatory disease characterized by the destruction of periodontal tissues and progressive loss of tooth-supporting structures. It is recognized as the leading cause of tooth loss in adults. Epidemiological data reveal that approximately 42% of adults over the age of 35 suffer from moderate periodontitis, 10% from severe forms, and nearly half of the adult population is affected by this non-communicable disease [1]. Although the exact pathogenesis of periodontitis remains incompletely understood, it is widely accepted that disease progression is driven by the disruption of the host–microbiome balance. This dysbiosis, often triggered by pathogenic biofilms, leads to a chronic inflammatory host response that ultimately damages periodontal tissues [2].

Oral biofilms are highly complex, and controlling their growth and elimination is challenging in periodontal therapy [3]. Conventional treatment strategies for periodontitis, including scaling and root planning, primarily aim to eliminate biofilms and restore tissue attachment [4]. However, given the complexity and resilience of oral biofilms, especially in subgingival regions, adjunctive therapies are often employed to enhance clinical outcomes [5]. Among these, systemic antibiotics have shown efficacy but are associated with systemic side effects and growing concerns over antibiotic resistance [6]. Low-level laser therapy (LLLT) has emerged as a non-invasive adjunctive option in periodontal therapy. Utilizing red or near-infrared light, LLLT is proposed to modulate biological processes such as cell proliferation, tissue regeneration, and inflammation [7]. Although several studies and meta-analyses have reported modest improvements in periodontal healing with LLLT, its efficacy and underlying mechanisms remain topics of ongoing investigation [8].

The antimicrobial potential of LLLT remains controversial. While some in vitro studies have demonstrated inhibitory effects on oral pathogens such as *Streptococcus mutans* and *Lactobacillus acidophilus* [9,10], others have reported minimal impacts on bacterial growth and the complex architecture of multispecies biofilms collected from periodontitis patients [11]. This raises critical questions regarding the true extent of LLLT’s antibacterial capabilities, particularly within structured subgingival biofilms that are highly resistant to conventional treatments.

In addition to microbial factors, host cellular responses play a pivotal role in the progression of periodontitis. The bacteria–host interaction initially activates immune and epithelial cells and, later, also engages gingival fibroblasts (GFs) and periodontal ligament cells (PDLs). These stromal cells, which meet the minimal criteria for mesenchymal stromal cells, play a pivotal role in maintaining periodontal tissue homeostasis, modulating immune responses, and contributing to tissue regeneration [12]. Both GFs and PDLs express various Toll-like receptors (TLRs) and are capable of responding to microbial stimuli by producing a range of pro-inflammatory mediators [12,13].

Previous studies have shown that LLLT can stimulate GF proliferation, reduce zoledronic acid-induced apoptosis [14,15,16], and enhance growth factor and collagen type I synthesis, which could benefit wound healing [16,17]. It also affects proliferation and osteogenic differentiation in PDLs [18,19].

Findings regarding the impact of LLLT on inflammation are inconsistent. Some studies have reported a reduction in lipopolysaccharide (LPS)-induced inflammatory responses in GFs and PDLs following exposure to diode (810 nm) and Nd:YAG (1064 nm) lasers [20,21], whereas others have reported both pro- and anti-inflammatory effects depending on laser parameters [19,22,23].

Given the current gaps in knowledge and the variability of reported outcomes, particularly with different wavelengths and energy settings, further research is needed to clarify the dual antimicrobial and immunomodulatory roles of LLLT, specifically for gingival and periodontal ligament cells.

In this study, we evaluated the effects of a diode laser emitting at 980 nm on the structure and viability of an in vitro subgingival seven-species periodontitis biofilm model. Additionally, we evaluated the anti-inflammatory potential of this laser in primary human gingival fibroblasts and periodontal ligament cells, with or without stimulation by *Porphyromonas gingivalis* LPS. Inflammatory responses were assessed by measuring the gene and protein expression levels of interleukin (IL)-6, IL-8, and monocyte chemoattractant protein (MCP)-1. To examine the impact of laser treatment on host cell viability, the MTT assay was employed. The 980 nm diode laser was selected due to its balanced absorption properties and moderate tissue penetration depth (2–3 mm), making it well suited for photobiomodulation applications [24]. LPS from *P. gingivalis* was used to mimic the inflammatory environment characteristic of periodontitis [25]. Special attention was given to the influence of different laser energy outputs and operational modes to better understand the therapeutic relevance and optimize the application parameters in periodontal treatment.

## 2. Results

### 2.1. Effect of LLLT on Seven-Species Biofilms

The impact of diode laser treatment on the composition and architecture of seven-species biofilms is presented in Figure 1 and Figure 2. Quantitative analysis revealed no significant differences in bacterial cell numbers between laser-treated samples and untreated controls (Figure 1). Confocal laser scanning microscopy (CLSM) showed consistent biofilm morphology across all groups, characterized by dense bacterial clusters interspersed with voids and channels (Figure 2). Overall, laser treatment had no discernible effect on biofilm composition or structural organization at clinical settings and even at higher settings.

### 2.2. Effect of LLLT on Gingival Fibroblasts and Periodontal Ligament Cells

#### 2.2.1. Cell Viability

Figure 3 illustrates the effect of diode laser treatment on the viability of GFs and PDLs, with and without *P. gingivalis* LPS stimulation. In the absence of LPS, laser treatment significantly reduced the viability of both GFs and PDLs across all power settings (0.1–0.5 W) and modes (continuous and pulsed). In the presence of LPS, cell viability was less affected by the laser; however, significant reductions were still observed at 0.3 W (pulsed mode) and 0.5 W (both modes) in GFs, and at 0.3 W (pulsed) and 0.5 W (all modes) in PDLs. *P. gingivalis* LPS alone (without laser treatment) showed a slight but inconsistently significant reduction in cell viability.

#### 2.2.2. Basal Inflammatory Response

Figure 4 shows the effects of LLLT on the basal gene and protein expression of IL-6, IL-8, and MCP-1. Gene expression levels remained unchanged across all treatment conditions. However, a general trend toward reduced cytokine secretion was observed at the protein level, particularly in PDLs. Significant reductions were noted for IL-6 (0.5 W, pulsed) and IL-8 (0.5 W, continuous) in GFs, and for MCP-1 at multiple settings. In PDLs, IL-6 secretion was significantly decreased at all settings, while IL-8 and MCP-1 were significantly reduced at specific powers and modes; i.e., IL-8 production was significantly decreased upon treatment with 0.1 W (pulsed) and 0.3 W (both modes) and MCP-1 production was significantly reduced at 0.1 W (pulsed) and at 0.3 W (continuous mode).

#### 2.2.3. Inflammatory Response to *P*. *gingivalis* LPS

As shown in Figure 5, LPS stimulation markedly increased the expression and secretion of IL-6, IL-8, and MCP-1 in both GFs and PDLs. Laser treatment modulated this response to varying degrees. In GFs, IL-6 gene expression was significantly reduced at 0.1 W (continuous) and 0.3 W (pulsed), while IL-8 expression showed mixed effects. MCP-1 expression remained unaffected. In PDLs, only IL-6 gene expression showed a significant decrease (0.1 W, pulsed). At the protein level, laser treatment in the presence of LPS generally reduced cytokine production. In GFs, significant reductions were observed for IL-6 (0.1 W, continuous) and for all mediators at 0.5 W (continuous). In PDLs, significant decreases were seen for IL-6 and IL-8 at 0.3 W (both modes), for MCP-1 at 0.5 W, and for IL-8 and MCP-1 at 0.1 W (pulsed).

#### 2.2.4. Mode Comparison

No consistent differences in any parameter were observed between the continuous and pulsed laser modes, regardless of the power setting or the presence of LPS.

## 3. Discussion

Low-level laser therapy (LLLT) has emerged as a promising adjunctive tool in non-surgical periodontal therapy due to its potential anti-inflammatory and antimicrobial properties. In this study, we assessed the impact of a 980 nm diode laser on both seven-species periodontal biofilms and the inflammatory response of primary human gingival fibroblasts (GFs) and periodontal ligament cells (PDLs).

Photobiomodulation using LLLT has been proposed as a novel approach in managing oral diseases, with several reports suggesting antibacterial effects [26]. To investigate this potential in a clinically relevant context, we employed a seven-species biofilm model [27] comprising commensal bacteria (*Streptococcus oralis*, *Streptococcus anginosus*, *Veillonella dispar*, *Actinomyces oris*), a bridging species (*Fusobacterium nucleatum*), and key periodontal pathogens (*P. gingivalis*, *Tannerella forsythia*). Despite the comprehensive nature of this model, LLLT at power outputs ranging from 0.3 to 2 W had no significant effect on biofilm composition or architecture (Figure 1 and Figure 2). To our knowledge, this is the first study to investigate the effects of LLLT on a complex, periodontitis-associated multispecies biofilm. Our findings are consistent with previous reports showing no impact of LLLT (850 nm) on the viability of planktonic *P. gingivalis* cells [28].

These results suggest that LLLT, with the parameters tested, does not exhibit direct antimicrobial effects against periodontal pathogens. This contrasts with findings in caries research, where LLLT (810–980 nm) has been shown to inhibit *S. mutans* and other cariogenic species such as *Lactobacillus casei*, *Actinomyces naeslundii*, and *L. Acidophilus* [10,29]. Additionally, LLLT (940 nm) has demonstrated bactericidal effects against Enterococcus faecalis, a species commonly implicated in endodontic infections [30]. Taken together, these results suggest that while LLLT may exert antimicrobial effects in certain oral disease contexts, its efficacy against periodontal biofilms remains unproven (this study), but its effectiveness should be tested in other biofilm models that include different periodontitis-associated species, particularly *Aggregatibacter actinomycetemcomitans*. The structural complexity, microbial diversity, and protective extracellular matrix of periodontal biofilms likely contribute to their resistance to laser irradiation even at settings higher than those used in the clinic. Therefore, future studies are needed to explore whether modifications in laser parameters (e.g., wavelength, energy density, exposure time) or combination strategies (e.g., with photosensitizers or mechanical debridement) might enhance the antibacterial potential of LLLT in periodontal therapy.

In the next phase of our study, we investigated the effects of LLLT on cell viability and the inflammatory response of GFs and PDLs, both in the absence and presence of *P. gingivalis* LPS. The inflammatory response was assessed by quantifying the gene and protein expression levels of IL-6, IL-8, and MCP-1. IL-6 plays a central role in orchestrating the host immune response and is closely associated with tissue destruction in periodontitis [31]. IL-8 and MCP-1 are critical chemokines involved in recruiting neutrophils and monocytes, respectively, to sites of infection or injury, thereby amplifying the local inflammatory response [32,33].

Our findings demonstrate that LLLT, particularly at higher power settings, can reduce cell viability in both GFs and PDLs up to 70%. Interestingly, this effect was more pronounced in the absence of LPS, suggesting that inflammatory priming may partially protect cells from laser-induced stress. Despite this reduction in viability, LLLT showed a modulatory effect on the inflammatory profile of these cells. In the absence of LPS, laser treatment tended to reduce the basal protein production of IL-6, IL-8, and MCP-1, with several treatment conditions leading to statistically significant decreases—especially in PDLs, which appeared more responsive to LLLT in this regard.

Our findings demonstrate that the inhibitory effect of LLLT on the viability of GFs and PDLs is less pronounced in the presence of *P. gingivalis* LPS. This suggests that inflammatory priming may confer a degree of resistance to LLLT-induced cytotoxicity. The slight reduction in cell viability caused by *P. gingivalis* LPS alone may also contribute to this diminished sensitivity. Significant viability reduction was observed only at higher power settings (0.5 W, both modes) and 0.3 W in pulsed mode, indicating a potential threshold effect of LLLT on cellular metabolism under inflammatory conditions. The mechanisms underlying the effects of LLLT on the viability of GFs and PDLs, both in the presence and absence of LPS, should be investigated in future studies by using specific apoptosis assays or by analyzing the expression of apoptosis-related proteins.

At the molecular level, LLLT had a minimal impact on the basal gene expression of IL-6, IL-8, and MCP-1 in both cell types. In the absence of LPS, LLLT treatment had no effect on the gene expression levels of all investigated immunomediators in GFs and PDLs (Figure 4A,B). However, a consistent trend toward reduced protein secretion of these mediators was observed following LLLT, although not all changes reached statistical significance. This aligns with prior research demonstrating modest, non-significant reductions in cytokine production following LLLT exposure (810 nm and 1064 nm) [20]. For instance, studies using GFs and PDLs have reported decreased basal levels of TNF-α and IL-1β upon LLLT treatment [19].

The inflammatory stimulation of GFs and PDLs with *P. gingivalis* LPS induced a robust upregulation of all measured pro-inflammatory mediators (Figure 5), consistent with previous findings [34]. LLLT only modestly attenuated the LPS-induced gene expression of IL-6 and IL-8, with no effect on MCP-1 transcripts. Interestingly, a slight upregulation of IL-6 gene expression was noted under one specific condition (0.3 W, pulsed mode), though its biological relevance appears limited.

More consistently, LLLT led to a significant reduction in LPS-induced protein production of IL-6, IL-8, and MCP-1 in both GFs and PDLs, although no clear pattern emerged regarding treatment power or mode. The observed discrepancies between gene and protein data may be attributed to temporal differences—while gene expression represents a snapshot at a single time point, protein accumulation in the culture medium reflects the cumulative response over the entire duration of LPS exposure. Additionally, reduced protein levels may partly result from the decreased metabolic activity of LLLT-treated cells. Finally, LLLT may influence post-transcriptional regulatory mechanisms, such as the modulation of microRNA expression [35].

Our results are supported by earlier studies showing that LLLT reduces LPS-induced cytokine production in oral cell types. For example, 810 nm and 1064 nm lasers were reported to inhibit IL-6, IL-8, and TNF-α production in GFs [20,21], while 660 nm LLLT suppressed similar cytokines in PDLs [23]. Importantly, our study extends this understanding to 980 nm LLLT and confirms that its anti-inflammatory effect is consistent across clinically relevant power settings and independent of whether the pulsed or continuous mode was applied.

Notably, the effects of LLLT on GFs and PDLs remain inconsistently reported in the literature. One study, in agreement with our findings, demonstrated that LLLT reduced GF proliferation without inducing cytotoxicity [36]. Conversely, other research employing similar laser settings found no significant impact on GF proliferation [37], while additional studies have even suggested that LLLT may enhance GF proliferation [14]. Similarly, studies on PDLs report variable outcomes, ranging from no observable effects on viability to increased viability following LLLT treatment [38,39,40]. These discrepancies likely arise from differences in experimental protocols, including laser parameters (wavelength, power, mode), exposure duration, and variations in cell culture conditions. Such variability underscores the importance of standardized methodologies to accurately assess the biological effects of LLLT on periodontal cell populations.

The absence of significant differences between the continuous-wave (CW) and pulsed-wave (PW) laser modes in our study aligns with previous findings. Al-Watban and Zhang reported comparable wound healing outcomes for CW and PW laser irradiation when the fluence was matched [41]. Similarly, Hashmi et al. concluded that although PW may offer theoretical advantages, such as a reduced thermal load and enhanced mitochondrial recovery, the biological effects of CW and PW tend to be similar when the total energy is controlled, particularly in in vitro settings [42]. These findings support the interpretation that, in our experiments, the fluence may have played a more critical role than the waveform itself.

From a clinical perspective, these findings are highly relevant. Periodontitis remains a leading cause of tooth loss and is characterized by persistent bacterial challenge and dysregulated inflammation. Standard non-surgical periodontal treatment aims to disrupt the biofilm to facilitate tissue healing, but recolonization often triggers renewed inflammation [43,44]. While adjunctive systemic antibiotics can be an option in preventing recolonization, the rising prevalence of antimicrobial resistance necessitates cautious use [45]. In this context, LLLT has been discussed as non-pharmacological adjunct to conventional therapy with at least short-term effectiveness [8]. Although it lacks direct antimicrobial efficacy against complex periodontal biofilms, its anti-inflammatory properties could reduce tissue destruction and modulate host responses upon bacterial recolonization. Additionally, LLLT has been reported to enhance regenerative mechanisms, including the stimulation of fibroblast and PDL proliferation, migration, and a reduction in oxidative stress [20,21,23]. These benefits further support the therapeutic potential of LLLT as part of an integrated periodontal treatment strategy.

Although the present study did not demonstrate significant antimicrobial activity for 980 nm LLLT alone, emerging evidence supports its potential when combined with adjunctive agents to enhance microbial control. As reviewed by Kikuchi et al., the antimicrobial efficacy of LLLT against periodontitis-associated bacteria may be augmented through the concurrent application of light-sensitive compounds such as methylene blue, toluidine blue O, malachite green, and indocyanine green [46]. Furthermore, the combined use of LLLT and Alveogyl has been shown to be more effective in managing alveolar osteitis than either treatment alone, implicating synergistic effects [47,48]. Additionally, the application of a 980 nm diode laser has been reported to promote wound healing and reduce postoperative complications in oral soft tissue procedures [49]. Collectively, these findings suggest that LLLT may enhance the efficacy of therapies targeting biofilm-associated infections when integrated into combination treatment protocols involving antimicrobial agents or photosensitizers.

A limitation of our study is the in vitro design. In each experiment, we focused on a single cell type, whereas in vivo processes involve interactions between multiple cell types. Additionally, we used bacterial LPS derived from a single bacterial species to stimulate the inflammatory response, whereas in vivo host cells are exposed to a variety of virulence factors from multiple bacterial species. Furthermore, we evaluated the effects of LLLT over a limited time frame without considering its long-term impact or potential cumulative effects.

## 4. Materials and Methods

### 4.1. Seven-Species Biofilm and Laser Treatment

A simplified subgingival biofilm model containing seven species, *Actinomyces oris* MG1 (OMZ 745), *Streptococcus oralis* SK248 (OMZ 607), *Streptococcus anginosus* ATCC 9895 (OMZ 871), *Streptococcus gordonii* (DSM6777), *Veillonella dispar* ATCC 17748 (OMZ 493), *F. nucleatum* ATCC 1095 (OMZ 598), *P. gingivalis* ATCC 33277 (OMZ 925), and *Tannerella forsythia* ATCC 43037, was developed based on Guggenheim et al. [50]. All bacteria were routinely cultivated anaerobically at 37 °C in brain–heart infusion (BHI) broth (Oxoid). *T. forsythia* cultures were supplemented with *N*-acetylmuramic acid and horse serum as described previously [51], while *V. dispar* required 0.1% sodium lactate (Sigma-Aldrich, St. Louis, MO, USA). For biofilm formation, individual bacterial cultures were adjusted to an OD_600_ of 1.0 and mixed in equal volumes. A 200 μL aliquot of this mixture was added to 1.6 mL of growth medium composed of 60% pooled human saliva, 10% fetal bovine serum (Sigma-Aldrich, St. Louis, MO, USA), and 30% modified fluid universal medium [52]. Biofilms were grown on sintered pellicle-coated hydroxyapatite (HA) discs, 9 mm in diameter (Clarkson Chromatography Products, South Williamsport, PA, USA), placed in 24-well tissue culture plates. The medium was refreshed at 16 and 24 h, and the discs were dip-washed twice daily with 0.9% NaCl.

After 64 h of anaerobic incubation at 37 °C, biofilms were dip-washed and treated with a 980 nm DenLase Diode Laser (China Daheng Group, Beijing, China). The laser was applied at a fixed distance of 0.5 cm, with a spot diameter of 8 mm (spot area: 0.503 cm^2^). Three irradiation settings were tested: (1) 0.3 W for 1 min (fluence: 35.8 J/cm^2^), (2) 0.3 W for 5 min (179 J/cm^2^), and (3) 2 W for 1 min (238.4 J/cm^2^). Untreated control discs were included in each experiment. Following treatment, discs were transferred to a fresh medium and incubated anaerobically for an additional 24 h with one additional wash step to evaluate the potential antimicrobial effect. Before analysis, biofilms were either harvested by vortexing in 0.9% NaCl for 2 min or fixed in 4% paraformaldehyde (Sigma-Aldrich) for 1 h at 4 °C for staining and confocal laser scanning microscopy.

### 4.2. Quantitative Analysis of Seven-Species Biofilm Composition by qPCR

Bacterial cell numbers within the seven-species biofilms were determined by quantitative PCR (qPCR) as described by Ammann et al. [53]. Genomic DNA was extracted from 500 µL of biofilm suspension using the GenElute™ Bacterial Genomic DNA Kit (Sigma-Aldrich, St. Louis, MO, USA). Species-specific primers targeting the 16S rRNA gene were used for amplification (Table 1). qPCR was performed using an MJ Mini Thermal Cycler (Bio-Rad, Vienna, Austria) to quantify the abundance of each bacterial species within the biofilm.

### 4.3. Structural Analysis of Seven-Species Biofilms by Confocal Laser Scanning Microscopy

To assess structural changes in the biofilms following laser treatment, fixed samples were stained with 15 μmol/L Sytox Green (Thermo Fisher Scientific, Waltham, MA, USA) and embedded in Mowiol [55]. Biofilm architecture was analyzed by CLSM using a Leica SP8 microscope (Vienna Institute of Biotechnology Imaging Center, University of Natural Resources and Life Sciences, Vienna, Austria). For each treatment group, three stained discs were imaged using a 40× objective. Images were processed and analyzed using software (ImageJ, version 1.53, National Institute of Health, Bethesda, MD, USA).

### 4.4. Cultivation of Human Cells

Primary human gingival fibroblasts (GFs) and periodontal ligament cells (PDLs) were isolated from the gingival tissue surrounding the third molars of eleven systemically healthy individuals undergoing tooth extraction for orthodontic reasons. The isolation of GFs and PDLs was approved by the Ethics Committee of the Medical University of Vienna (Protocol No. 1079/2019, valid until 18 March 2026). Written informed consent was obtained from all donors prior to sample collection. The study was conducted in accordance with the Declaration of Helsinki and the Medical University of Vienna’s guidelines for Good Scientific Practice. None of the donors were receiving medication at the time of extraction. Cells were obtained using the outgrowth method as described previously [34] and cultured in Dulbecco’s Modified Eagle Medium (DMEM, Sigma-Aldrich, St. Louis, MO, USA) supplemented with 10% fetal bovine serum (FBS Xtra, Capricorn Scientific GmbH, Ebsdorfergrund, Germany), 100 U/mL penicillin, and 100 μg/mL streptomycin (Sigma-Aldrich, St. Louis, MO, USA). Cultures were maintained at 37 °C in a humidified atmosphere with 5% CO_2_. The medium was replaced every three days, and cells were passaged upon reaching 80–90% confluence. Experiments were conducted using cells from passages 4 to 7. To consider inter-individual variability inherent to primary cells [56], cells of at least four different donors were used per experiment.

### 4.5. Treatment of Human Cells with Laser and P. gingivalis LPS

GFs and PDLs were seeded in 24-well plates at a density of 5 × 10^4^ cells per well in 500 µL of DMEM supplemented with 10% FBS and antibiotics. Cells were seeded in a checkerboard pattern, leaving adjacent wells empty to prevent laser interference during treatment.

After 24 h, the medium was removed, and cells were rinsed once with phosphate-buffered saline (PBS). Cells were then exposed to 980 nm diode laser light (Denlace, China Daheng Group, Inc., Beijing, China). The laser was applied at a fixed distance of 0.5 cm for 60 s in either the continuous-wave (CW) or pulsed-wave (PW) mode, using power settings of 0.1 W, 0.3 W, or 0.5 W. Based on a laser spot diameter of 8 mm (corresponding to an area of 0.503 cm^2^), the resulting fluences were calculated as 11.9 J/cm^2^, 35.8 J/cm^2^, and 59.6 J/cm^2^, respectively. Under these conditions, the laser beam diameter at the well surface was 8 mm. Laser irradiation was performed without medium or PBS in the wells. Immediately afterwards, 500 µL of fresh DMEM with FBS and antibiotics was added (see above). Control wells underwent identical handling without laser exposure.

Following treatment, cells were incubated for 2 h before the medium was replaced with DMEM containing antibiotics but no FBS. In selected wells, this medium was supplemented with 1 µg/mL *P. gingivalis* LPS (ATCC 33277, standard, InvivoGen, San Diego, CA, USA) and 250 ng/mL soluble CD14 (sCD14, Sigma-Aldrich, St. Louis, MO, USA), based on previously published protocols [34]. Cells were cultured under these conditions for 24 h prior to downstream analysis.

### 4.6. Cell Viability and Proliferation Assay

Cell viability was assessed using the 3-(4,5-Dimethylthiazol-2-yl)-2,5-diphenyltetrazoliumbromid (MTT) assay, which measures mitochondrial metabolic activity as an indicator of viable, proliferating cells [57]. Briefly, 100 µL of MTT solution (5 mg/mL in PBS, Sigma-Aldrich, St. Louis, MO, USA) was added to each well and incubated for 2 h at 37 °C. After incubation, the supernatant was removed, and 500 µL of dimethyl sulfoxide (Merck, Darmstadt, Germany) was added to solubilize the formazan crystals. Following a 5 min incubation, 100 µL of the resulting solution was transferred to a 96-well plate. Optical density (OD) was measured at 570 nm using a microplate reader (Synergy HTX, BioTek, Winooski, VT, USA). All measurements were performed in quadruplicate.

### 4.7. Gene Expression and Protein Production Analysis

Gene expression was assessed by qPCR using the TaqMan^TM^ Gene Expression Cells-to-Ct Kit (Thermo Fisher Scientific, Waltham, MA, USA) as previously described [34,58]. Cells were lysed immediately after treatment, and lysates were stored at −20 °C until further processing. The reverse transcription of lysates to cDNA was performed using a Primus 96 advanced thermocycler (Applied Biosystems, Foster City, CA, USA) under the following conditions: 37 °C for 60 min, 95 °C for 5 min, followed by cooling to 4 °C. qPCR was carried out using an ABI StepOnePlus device (Applied Biosystems, Foster City, CA, USA) with the following thermal profile: 95 °C for 10 min, followed by 50 cycles of 95 °C for 15 s and 60 °C for 1 min. TaqMan^®^ Gene Expression Assays (Applied Biosystems, Foster City, CA, USA) used were as follows: GAPDH (Hs99999905_m1); MCP-1 (Hs00234140_m1); IL-8 (Hs00174103_m1); and IL-6 (Hs00985639_m1). Gene expression changes were calculated using the 2^−ΔΔCt^ method, taking GAPDH as the housekeeping gene and cells without any treatment as a control.

The concentration of the inflammatory mediators IL-6, IL-8, and MCP-1 in conditioned media was determined using uncoated ELISA kits (Thermo Fisher Scientific, Waltham, MA, USA) according to the manufacturer’s protocol. The sensitivity of the ELISA was 2 pg/mL for IL-6 and IL-8 and 7 pg/mL for MCP-1.

### 4.8. Statistical Analysis

Data are presented as the mean ± standard error of the mean (s.e.m.) from at least four biological replicates. Due to the limited sample size, non-parametric statistical analysis was applied. Differences between treatment groups were evaluated using the Wilcoxon signed-rank test. All analyses were performed using statistics software (SPSS 28.0, IBM, Armonk, NY, USA).

## 5. Conclusions

In summary, our study reinforces the concept that LLLT exerts beneficial immunomodulatory effects on periodontal cells, even under strong bacterial stimulation. While LLLT does not appear to directly disrupt pathogenic biofilms, its ability to attenuate excessive inflammatory responses and potentially support regenerative processes highlights its value as an adjunct modality in the management of periodontal disease.

## Figures and Tables

**Figure 1 ijms-26-06803-f001:**
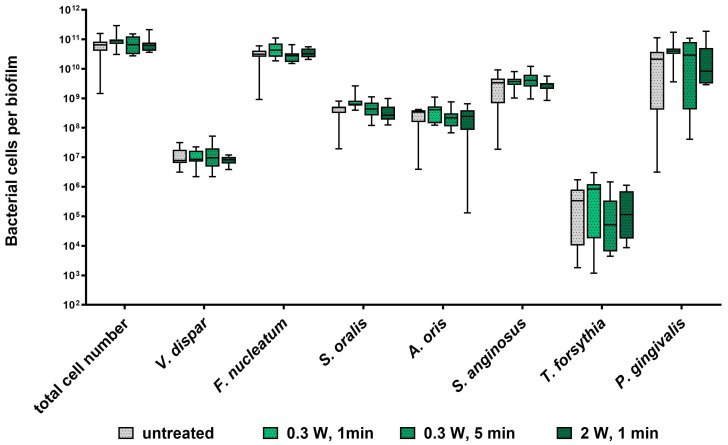
Impact of diode laser treatment on bacterial cell numbers in a subgingival seven-species biofilm. Whisker’s boxplots (5th to 95th centile) showing cell numbers of all species determined by qRT-PCR from three independent experiments and four replicates, each (N = 12).

**Figure 2 ijms-26-06803-f002:**
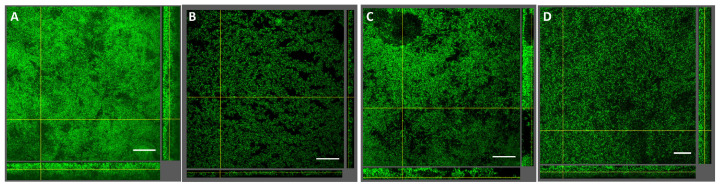
Impact of diode laser treatment on seven-species biofilm structure. Confocal images of biofilms stained with Sytox Green are shown. (**A**) Untreated control, (**B**) 0.3 W, 1 min, (**C**) 0.3 W, 5 min, (**D**) 2.0 W, 1 min. A representative area for one disc is shown, with a top view in the large panel and a side view with the biofilm–disc interface directed towards the top view. Scale bar: 40 μm, 40× objective, 0.75× zoom.

**Figure 3 ijms-26-06803-f003:**
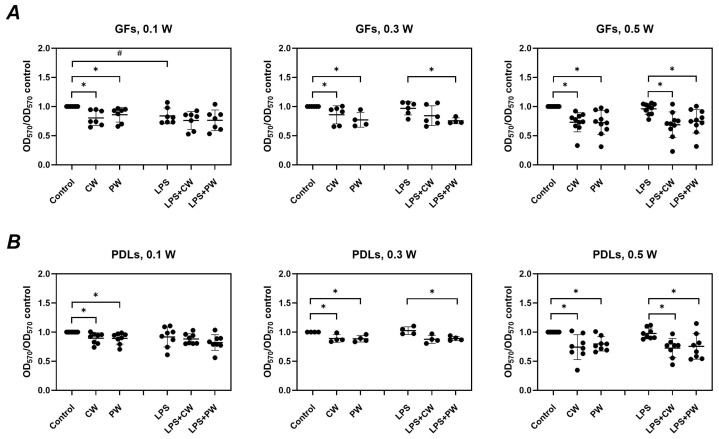
Impact of diode laser treatment on the viability of gingival fibroblasts (GFs) and periodontal ligament cells (PDLs) under different conditions. GFs (**A**) and PDLs (**B**) were treated with a diode laser with either a continuous wave (CW) or a pulsed wave (PW) with a power of 0.1, 0.3, or 0.5 W. The cells were further cultured for an additional 24 h in the absence or presence of 1 µg/mL of *P. gingivalis* lipopolysaccharide (LPS) and 250 ng/ml of soluble CD14, and cell viability was measured using the 3-(4,5-dimethylthiazol-2-yl)-2,5-diphenyltetrazolium bromide (MTT) method. Y-axes show the optical density values measured at 570 nm (OD_570_) normalized to those of untreated control cultured without LPS. Each point represents the data of cells of one donor measured in technical triplicate, and the bars show the mean value ± standard deviation. *—significantly lower viability upon treatment with the diode laser; #—significantly lower viability after culturing with LPS.

**Figure 4 ijms-26-06803-f004:**
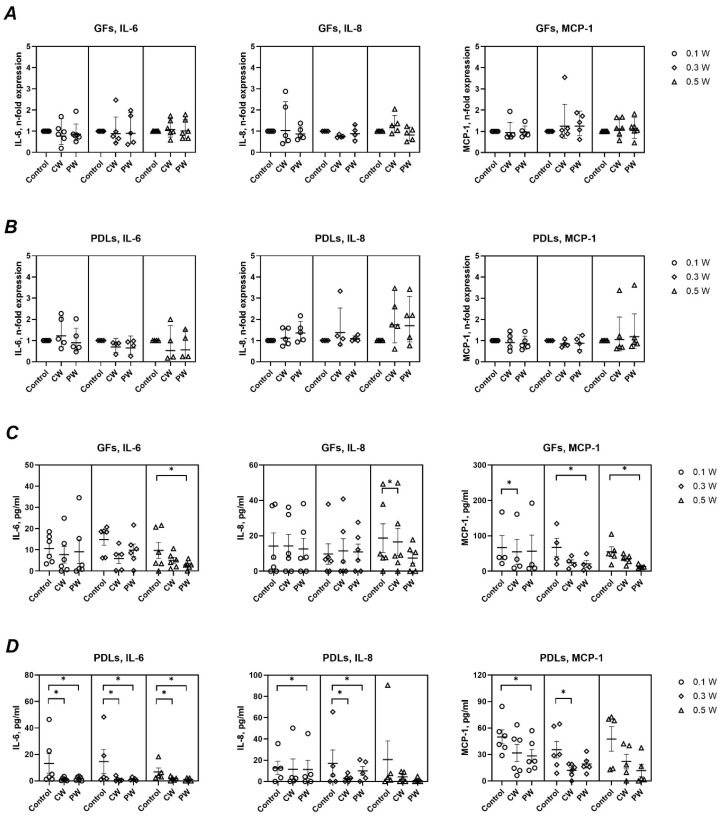
Impact of different diode laser treatments on basal gene expression and protein production of IL-6, IL-8, and MCP-1 by gingival fibroblasts (GFs) and periodontal ligament cells (PDLs). GFs (**A**,**C**) and PDLs (**B**,**D**) were treated with a diode laser with either a continuous wave (CW) or a pulsed wave (PW) with a power of 0.1, 0.3, or 0.5 W. The cells were further cultured for an additional 24 h, and the gene expression of interleukin (IL)-6, IL-8, and monocyte chemoattractant protein (MCP)-1 (**A**,**B**) and the content of corresponding proteins in conditioned media (**C**,**D**) were measured by qPCR and ELISA, respectively. Y-axes (**A**,**B**) show the n-fold gene expression in relation to the untreated control (no laser) and calculated by the 2^−ΔΔCt^ method. Each point represents the data of cells of one donor measured in technical duplicates. Y-axes (**C**,**D**) show the concentration of corresponding proteins measured by ELISA. The bars show mean values ± s.e.m; *—significantly lower gene expression or protein production upon treatment with the diode laser.

**Figure 5 ijms-26-06803-f005:**
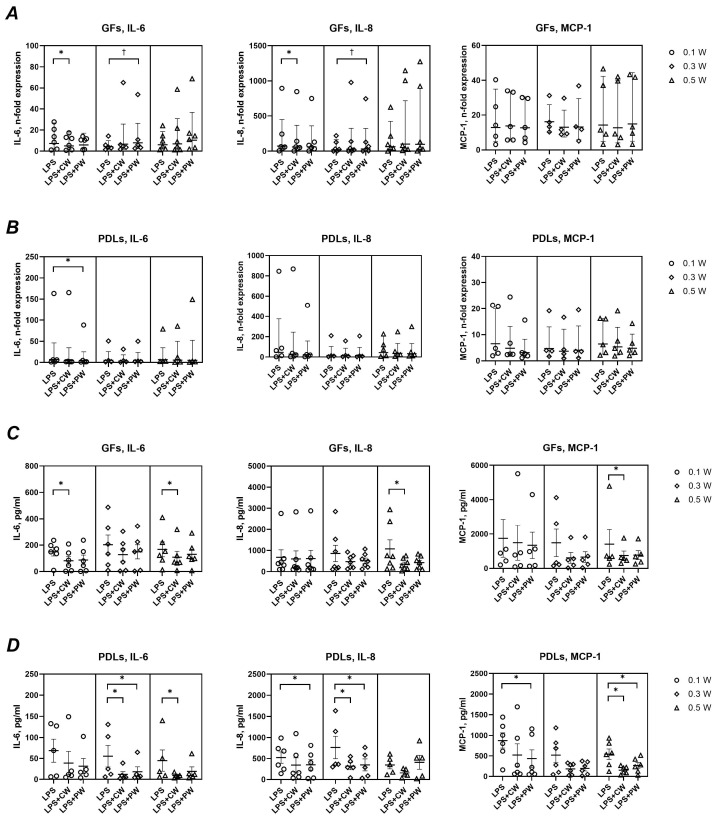
Impact of different diode laser treatments on *P. gingivalis* LPS-induced gene expression and protein production of interleukin (IL)-6, IL-8, and monocyte chemoattractant protein (MCP)-1 by gingival fibroblasts (GFs) and periodontal ligament cells (PDLs). GFs (**A**,**C**) and PDLs (**B**,**D**) were treated with a diode laser with either a continuous wave (CW) or a pulsed wave (PW) with a power of 0.1, 0.3, or 0.5 W. The cells were further cultured for an additional 24 h in the presence of 1 µg/ml of *P. gingivalis* lipopolysaccharide (LPS) and 250 ng/mL soluble CD14, and the gene expression of IL-6, IL-8, and MCP-1 (**A**,**B**) and the content of corresponding proteins in conditioned media (**C**,**D**) were measured by qPCR and ELISA, respectively. Y-axes (**A**,**B**) show the n-fold gene expression in relation to untreated control (no laser) and calculated by the 2^−ΔΔCt^ method. Each point represents the data of cells of one donor measured in technical duplicates. Y-axes (**C**,**D**) show the concentration of corresponding proteins measured by ELISA. The bars show mean values ± s.e.m; *—significantly lower gene expression or protein production upon treatment with the diode laser. †—significantly higher gene expression upon treatment with the diode laser.

**Table 1 ijms-26-06803-t001:** Primers used for quantitative PCR.

Organism	Sequence (5′-3′)	Reference
*S. oralis*	ACCAGGTCTTGACATCCCTCTGACC	[54]
	ACCACCTGTCACCTCTGTCCCG	
*S. anginosus*	ACCAGGTCTTGACATCCCGATGCTA	[54]
	CCATGCACCACCTGTCACCGA	
*A. oris*	GCCTGTCCCTTTGTGGGTGGG	[54]
	GCGGCTGCTGGCACGTAGTT	
*V. dispar*	CCCGGGCCTTGTACACACCG	[54]
	CCCACCGGCTTTGGGCACTT	
*F. nucleatum*	CGCCCGTCACACCACGAGA	[54]
	ACACCCTCGGAACATCCCTCCTTAC	
*T. forsythia*	CGATGATACGCGAGGAACCTTACCC	[54]
	CCGAAGGGAAGAAAGCTCTCACTCT	
*P. gingivalis*	GCGAGAGCCTGAACCAGCCA	[54]
	ACTCGTATCGCCCGTTATTCCCGTA	

## Data Availability

The data presented in this study are available on request from the corresponding author.

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
