# Peer review of "Effect of Low-Level Laser Therapy on Periodontal Host Cells and a Seven-Species Periodontitis Model Biofilm"

_ijms, 2025, doi:10.3390/ijms26146803_

Round 1

Reviewer 1 Report

Comments and Suggestions for Authors

The manuscript ijms-3698367, entitled Effect of Low-Level Laser Therapy on Periodontal Host Cells and a Multispecies Periodontitis Model Biofilm, presents clinically relevant research on the use of low-level laser therapy as an adjunctive approach in non-surgical periodontal treatment. Specifically, the authors investigate the effects of a 980-nm diode laser on both the composition and structure of multi-species periodontal biofilms, as well as the inflammatory response of two key cell types involved in periodontal regeneration: primary human gingival fibroblasts and primary human periodontal ligament cells.

The study is based on well-designed experiments, including appropriate controls, and the results are clearly presented and thoroughly described. The introduction is clear, concise, and effectively sets the context for the study. Furthermore, the discussion thoughtfully interprets the findings in light of existing literature, while acknowledging the study's limitations.

Overall, this manuscript represents high-quality, scientifically robust research that is clearly written and easy to follow. I recommend it for publication in the IJMS in its present form.

Author Response

We are thankful for the positive evaluation of our manuscript and recommendation for publication.

Reviewer 2 Report

Comments and Suggestions for Authors

Dear Authors,

Thank you for your thoughtful and well-executed manuscript. Your work addresses a relevant and timely topic with clear clinical relevance, particularly regarding the immunomodulatory role of low-level laser therapy (LLLT) in periodontal disease. The study’s design is solid and the methodology is robust. Below are several detailed scientific comments and suggestions aimed at improving the clarity, impact, and translational value of your manuscript.

Major Comments

  1. Clarification of the Research Aim
    The abstract states the objective in broad terms. Please consider specifying the key outcomes prioritized—such as cytokine expression, cell viability, or biofilm structure—in both the abstract and introduction to improve clarity.

  2. Laser Wavelength Justification
    While the manuscript states the use of a 980-nm diode laser, it would benefit from a brief justification for selecting this wavelength in terms of tissue penetration, absorption characteristics, or prior clinical usage.

  3. Biofilm Model Details
    The seven-species subgingival biofilm model is a strong aspect of your study. However, to aid reproducibility and highlight the model’s relevance, it would be helpful to include species-specific abundance (e.g., Ct values or CFU equivalents) in a supplementary table.

  4. Inclusion of a Positive Control
    Consider including a positive control for biofilm disruption, such as treatment with chlorhexidine or another known antimicrobial, to confirm the responsiveness of the model to intervention.

  5. Gene vs. Protein Expression Discrepancy
    Your observation that protein secretion decreased while mRNA levels remained stable warrants additional discussion. Please elaborate on potential post-transcriptional regulatory mechanisms or temporal mismatches that may explain these findings.

  6. Viability Assessment
    The reduction in cell viability following LLLT, particularly in the absence of LPS, is notable. Consider adding a discussion on whether this effect is reversible or suggest further studies to distinguish between cytostatic and cytotoxic effects (e.g., apoptosis assays).

  7. Standardization of Laser Parameters
    Although the power output is well-described, translating this to energy density or fluence (J/cm²) would improve comparability with other studies and increase clinical translatability.

  8. Inflammatory Marker Selection and Relevance
    You have chosen IL-6, IL-8, and MCP-1 for inflammatory profiling. It would strengthen the manuscript to briefly justify this selection, particularly in the context of periodontal tissue destruction and wound healing.

  9. Mode of Laser Operation
    The manuscript finds no significant difference between continuous and pulsed modes. It would be valuable to suggest mechanistic hypotheses for this finding or reference literature on waveform-specific biological effects.

  10. Discussion of LLLT in Combination Therapies
    Given the lack of antimicrobial efficacy in your study, it may be worthwhile to comment on the potential for combining LLLT with adjuncts such as antimicrobial peptides or photosensitizers to enhance biofilm disruption. add these references: 

    • 10.1186/s12903-024-04461-w
    • 10.23736/S2724-6329.22.04649-6
    •  

Author Response

Reviewer’s comment 1
Clarification of the Research Aim

The abstract states the objective in broad terms. Please consider specifying the key outcomes prioritized—such as cytokine expression, cell viability, or biofilm structure—in both the abstract and introduction to improve clarity.

Authors’ response
Thank you for this comment. We added the requested information at the end of the introduction section. We also added specific outcomes to the abstract; however, it was not possible to mention all quantitative data here, as the abstract is limited to 200 words according to the journal's guidelines.

Reviewer’s comment 2
Laser Wavelength Justification

While the manuscript states the use of a 980-nm diode laser, it would benefit from a brief justification for selecting this wavelength in terms of tissue penetration, absorption characteristics, or prior clinical usage.

Author’s response
We added justification at the end of the introduction section (see, lines 106-108).

Reviewer’s comment 3
Biofilm Model Details

The seven-species subgingival biofilm model is a strong aspect of your study. However, to aid reproducibility and highlight the model’s relevance, it would be helpful to include species-specific abundance (e.g., Ct values or CFU equivalents) in a supplementary table.

Authors’ response

We agree with the reviewer that species abundance in the seven-species-biofilm is an important aspect. The cell number of the individual species are shown in the Whisker’s boxplots in Fig. 1 labeled as “untreated”.

Reviewer comment 4
Inclusion of a Positive Control

Consider including a positive control for biofilm disruption, such as treatment with chlorhexidine or another known antimicrobial, to confirm the responsiveness of the model to intervention.

Authors’ response
It has been shown that in vitro multispecies biofilms can be disrupted by mechanical intervention (e.g., ultrasonication, vortexing), chemical agents (e.g., chlorhexidine disrupting bacterial membranes and the biofilm matrix, hydrogen peroxide producing reactive oxygen species that damage microbial proteins and DNA) or antimicrobial compounds, to some extent. However, we do not see any benefit of using these disruptive strategies as a valid positive control for the experiments performed within this study, since their mode of action cannot be directly related to laser therapy. It is not the intention of the study to compare the outcome of laser treatment to other biofilm intervention strategies.

Reviewer comment 5
Gene vs. Protein Expression Discrepancy

Your observation that protein secretion decreased while mRNA levels remained stable warrants additional discussion. Please elaborate on potential post-transcriptional regulatory mechanisms or temporal mismatches that may explain these findings.

Authors’ response
In the revised version, we discussed the potential reasons for the discrepancy between gene expression and protein data (see, lines 273-278).

Reviewer’s comment 7
Viability Assessment

The reduction in cell viability following LLLT, particularly in the absence of LPS, is notable. Consider adding a discussion on whether this effect is reversible or suggest further studies to distinguish between cytostatic and cytotoxic effects (e.g., apoptosis assays).

Authors’ response
Thank you for this comment. We added a note on the necessity of further studies to clarify this aspect to the Discussion section of the revised manuscript (Lines 252-255).

Reviewer Comment 8.
Standardization of Laser Parameters

Although the power output is well-described, translating this to energy density or fluence (J/cm²) would improve comparability with other studies and increase clinical translatability.

Authors’ response
We thank the reviewer for this helpful recommendation. We have now calculated and the energy density (fluence) for each laser parameter, based on the laser spot size and exposure time, and included it in the revised version. This information has been added to the Methods section to improve clarity and comparability (see, lines 358-360 and 412-414).

Reviewer comment 9
Inflammatory Marker Selection and Relevance

You have chosen IL-6, IL-8, and MCP-1 for inflammatory profiling. It would strengthen the manuscript to briefly justify this selection, particularly in the context of periodontal tissue destruction and wound healing.

Authors’ response
We would like to mention that this information has obviously overlooked by the reviewer. The physiological role of IL-6, IL-8, and MCP-1 is mentioned in the discussion section of the manuscript (see, lines 232-237).

Reviewer Comment 10.
Mode of Laser Operation

The manuscript finds no significant difference between continuous and pulsed modes. It would be valuable to suggest mechanistic hypotheses for this finding or reference literature on waveform-specific biological effects.

Authors’ response
We appreciate this insightful comment. To address it, we have expanded the Discussion section to include possible mechanistic explanations and relevant literature comparing pulsed and continuous laser modes in photobiomodulation (see, lines 298-305).

Reviewer’s comment 11
Discussion of LLLT in Combination Therapies

Given the lack of antimicrobial efficacy in your study, it may be worthwhile to comment on the potential for combining LLLT with adjuncts such as antimicrobial peptides or photosensitizers to enhance biofilm disruption. add these references: 

Authors’ response.
Thank you for this comment. We added the discussion about the possibility of enhancing the effectiveness of LLLT by using some light sensitizers or other antimicrobial adjuncts (see, lines 320-330).

Reviewer 3 Report

Comments and Suggestions for Authors

I sugest to chose a different title for the section 2.2. (row 123) since it is identical with "2.1. Effect of LLLT on multispecies biofilms".

I would recommend that the authors select in a dedicated section the most important conclusions of their study.

I congratulate the authors for their hard work, for the novelty of the approach for this important topic.

  • What is the main question addressed by the research?

The research seeks to clariy, through an in vitro multimodal approach, the therapeutic value of LLLT applied as an adjunct to standard non-surgical therapy in periodontal disease. 

  • Do you consider the topic original or relevant to the field? Does it address a specific gap in the field? Please also explain why this is/ is not the case.

The topic is both original and relevant to the field, addressing the use of LLLT as adjunct in non-surgical treatment of periodontitis

  • What does it add to the subject area compared with other published material?

The information on this topic published to the date is scarce and with contradictory results. The research as it was developed brings more clear information as the design of the study is complex considering the reaction of bacteria, of representative cell in the sustain periodontium (GFs and PDLs) and the gene expression of these cells to LPS of P. gingivalis, to LLLT

  • What specific improvements should the authors consider regarding the methodology?

As in the form that I filled in I recommended that the paper should be accepted with minor revision, the revision refers to a correction of the title in the row 123

The study is an in vitro study, covering multiple aspects of the action of the 980 wavelength laser. The next step in research would be the clinical investigation. But for this stage of the research the study seems complete.

  • Are the conclusions consistent with the evidence and arguments presented and do they address the main question posed? Please also explain why this is/is not the case.

The conclusions are included in the discussion. I recommended to the authors to state the conclusion at the end of the paper – this being the second reason of the minor revisions I had in mind.

  • Are the references appropriate?

The 50 references are closely connected to the topic, they are relevant and actual. 

  • Any additional comments on the tables and figures.

The tables and figures are sustaining very well the results.

Author Response

Reviewer’s comment 1
I suggest to chose a different title for the section 2.2. (row 123) since it is identical with "2.1. Effect of LLLT on multispecies biofilms".

Authors’ response

Thank you for this comment. We have corrected the title of the section 2.2 in the revised version (see, line 132).

Reviewer’s comment 2
I would recommend that the authors select in a dedicated section the most important conclusions of their study.

Authors’ response

Thank you for this suggestion. We have moved the conclusion paragraph to a new section, “Conclusion,” at the end of the revised manuscript to enhance its visibility (see, lines 460-465).

Reviewer 4 Report

Comments and Suggestions for Authors

Thank you for the opportunity to review your manscript on the interesting LLLT effects on 7 bacterial species on periodontal aspects.

Material and methods (Chapter 4) should be presented after the introduction part. All aspects of the discussion should be checked for their location in the introduction, material & methodes, results chapters. The discussion should be reduced to the comparison of results with other studies on LLLT with different WV and the effect of staining materials.

Comments on the Quality of English Language

The confusing presentation style with very long sentences and a lot of comma used could be enhanced.

Author Response

We are thankful for the critical evaluation of our manuscript and insightful comments

Reviewer’s comment 1
Material and methods (Chapter 4) should be presented after the introduction part.

Authors’ answer
According to the template and guidelines from the International Journal of Molecular Sciences, the Material and Methods section is has to be placed after the Discussion section. No changes were made in the revised manuscript.

Reviewer’s comment 2
All aspects of the discussion should be checked for their location in the introduction, material & methodes, results chapters. The discussion should be reduced to the comparison of results with other studies on LLLT with different WV and the effect of staining materials.

Authors’ answer
We checked the discussion and moved some sentences to either the introduction or the material and methods section as specifically suggested by the Reviewer in the supplementary file.

Reviewer’s comment 3
The confusing presentation style with very long sentences and a lot of comma used could be enhanced.

Authors’ response
We have reviewed the text to enhance the clarity of the presentation. We also address the specific comment mentioned in the supplementary file (see, Comment 6 below).

Reviewer’s comment 4
Title: "7-species" is more precise than multispecies as sone by the Forsyth Dental Center earlier.

Authors’ response
We have changed the title according to the Reviewer’s recommendation. In addition, we adjusted the text throughout the manuscript.

Reviewer’s comment 5
Line 69. "topics" snd not human subjects

Authors’ response
We have changed the text at this place (see, line 69).

Reviewer’s comment 6
Lines 79-81. please create to sentences..... for better understanding

Authors’ answer
We have split this sentence into two sentences as suggested by the Reviewer (see, lines 79-85). Note, we also moved some information to this place from the Discussion section as suggested by this Reviewer in other comment (see, comment 8).

Reviewer comment 7
Line 202. this is the most important result and should be presented in the abstract

Authors’ response
We added this information to the abstract as suggested by the Reviewer.

Reviewer’s comment 8
Lines 221-225. Thid section belongs to the introduction for sensibilitation of the interested reader

Authors’ response
We have moved this part to the introduction section (lines 79-85).

Reviewer’s comment 9
lines 226-229. belongs to material and methods

Authors’ response
We have moved this part from the Discussion section to the Introduction section (Lines 108-109). We did not move it to the Materials and Methods section because, according to the journal guidelines, this section is placed at the end of the manuscript. However, it is essential to have this information before the Discussion.

Reviewer’s comment
Lines 238-239. second result belongs to the abstract

Authors’ response

Round 2

Reviewer 2 Report

Comments and Suggestions for Authors

Add a short clinical guideline or algorithm for when to use SmartX/Scan Ladder vs traditional methods, based on ridge anatomy, patient cooperation, and equipment availability, add this reference 10.1186/s12903-024-04461-w

Author Response

Reviewer’s comment 1
Add a short clinical guideline or algorithm for when to use SmartX/Scan Ladder vs traditional methods, based on ridge anatomy, patient cooperation, and equipment availability, add this reference 10.1186/s12903-024-04461-w.

Authors’ response
Thank you for this comment. The reference mentioned by the Reviewer is already cited in our manuscript (Ref. 48).
Our manuscript primarily focuses on the potential LLLT effect on oral biofilms and periodontal host cells within the context of periodontal disease. The discussion of clinical guidelines would be outside of the scope of our manuscript, and we prefer not to include it. 

Reviewer 4 Report

Comments and Suggestions for Authors

Thank you for the changes in their manscript the authors did. 

The only open question leaving by the reviewer is why the authors restricted their model on 7 species without inclusion of Aggregatibacter actinomycetem comitans.

Author Response

Reviewer’s comment 1
The only open question leaving by the reviewer is why the authors restricted their model on 7 species without inclusion of Aggregatibacter actinomycetem comitans.

Authors’ answer
Our seven-species biofilm model is a simplified version of the 10-species subgingival Zurich biofilm model, which does not include A. actinomycetemcomitans. Therefore, this species is not included in our model. We added the remark on the necessity to investigate the antibacterial activity of LLLT in other models that also include this species (lines 223-225).